# Shape-Guided Anatomy Segmentation with Mesh Prompts

**Dingjie Su**[1]                                                    DINGJIE.SU@VANDERBILT.EDU
**Yihao Liu**[2]                                                     YIHAO.LIU@VANDERBILT.EDU
**Lianrui Zuo**[2]                                                   LIANRUI.ZUO@VANDERBILT.EDU
**Benoit M. Dawant**[1,2]                                          BENOIT.DAWANT@VANDERBILT.EDU
[1] *Vanderbilt University Department of Computer Science, Nashville, TN, USA*
[2] *Vanderbilt University Department of Electrical and Computer Engineering, Nashville, TN, USA*

**Editors:** Accepted for publication at MIDL 2025

## Abstract

We present a novel technique for segmenting anatomical structures in medical images by using a canonical mesh as a prompt for the structure to be segmented. Unlike point-prompted segmentation methods, such as those based on Segment-Anything Models, mesh prompting reduces the ambiguity associated with point prompts and provides a stronger shape prior, which is particularly advantageous for many medical applications. Our approach performs mesh-prompted segmentation by registering the signed distance function (SDF) of the mesh to the target image using a vector-field attention network trained with boundary-based loss terms. Before registration, the prompted mesh is roughly aligned with the structure in the target image using a center prompt provided by the user. This method allows for independent initialization of each structure's position and the prediction of deformation fields specific to each structure, which offers advantages over segmentation via direct image registration that typically relies on a single deformation field to accommodate all structures. Additionally, it preserves surface correspondence better than image registration using region-based loss terms. We evaluate our method on two CT datasets featuring common ear and body structures. A comparison of our technique with image registration and other state-of-the-art segmentation methods shows that our approach achieves superior segmentation accuracy. We open source our code at https://github.com/MIP-Lab/mesh-prompted-segmentation.git

**Keywords:** image registration, anatomy segmentation, image correspondence building

## 1. Introduction

Vision foundation models have recently gained significant attention due to their high generalizability and adaptability. In image segmentation, the Segment-Anything Model (SAM) (Kirillov et al., 2023) stands out as a prominent foundation model. SAM introduces a novel segmentation task that segments target structures using prompts such as points, bounding boxes, scrambled masks. This task, which we referred to as point-prompted segmentation (PPS) [1], enables SAM to adapt to new segmentation tasks with minimal or no additional training. Consequently, SAM has inspired numerous studies in the medical field to leverage or extend its capabilities for medical image segmentation (Lei et al., 2024; Dong et al., 2024;

---

1. We regard bounding boxes also as point prompts because they are represented as corner points and handled similarly as point prompts in SAM. Scrambled masks are not considered in this paper because they are less common in practice due to high complexity.

Chen et al., 2024; Ali et al., 2023; Ma et al., 2024; Huang et al., 2024; Zhu et al., 2024), which we collective name as MedSAM (SAM for medical images).

An important application of SAM in the medical field is adapting it to specific medical datasets through fine-tuning. While this approach has proven effective in many applications, we argue that its utility remains constrained by the PPS task on which SAM is based. This constraint leads to suboptimal segmentation performance when faced with common challenges in medical image segmentation, such as image noise and blurriness, overlapping or intertwined structures, and implicit boundaries. In these cases, the interior points or bounding boxes within the PPS task become ambiguous to the model, often resulting in segmentation failure. Fig. 1 left illustrates some of these challenges with an example of four ear structures: modiolus (MD), scala tympani (ST), scala vestibuli (SV), and the labyrinth. Here, ST and SV are intertwined and both overlap with the labyrinth, and the boundary between ST and SV is often indistinct in the CT image. While increasing the number of prompts may help alleviate these issues, accurately localizing numerous points in the test images becomes a challenge. Similar challenges have been highlighted in recent studies (Huang et al., 2024; Dong et al., 2024), but no satisfactory solutions have been proposed. As a result, SAM falls short of its "segment anything" promise in these scenarios.

This paper aims to develop segmentation techniques for anatomical structures that address the previously mentioned limitations and have the potential to adapt to unseen structures when supported by large-scale pretraining (as is done with SAM). Our insight is that the core limitation with PPS lies in the inadequate specification of target structures. Recognizing that anatomical structures often exhibit predictable shapes, we propose mesh-prompted segmentation (MPS) for providing a much more detailed specification of the target structure without extra burden for the user. Instead of point prompts, we condition the segmentation on a canonical surface mesh representing the structure of interest. Unlike the randomly distributed point prompts, mesh points collectively form meaningful shapes that the model must respect. However, the MPS task cannot be directly addressed by SAM's current architecture. Therefore, new techniques are needed to integrate the holistic representation of the mesh.

Inspired by registration-based image segmentation, we implemented MPS by learning the registration between the mesh and the image. To start with, the surface mesh of an anatomical structures is obtained and converted into discretized signed distance function (SDF), which represents the mesh as a 3D volume termed as signed distance map (SDM). Each voxel in the SDF contains the distance to the nearest boundary point of the mesh, with the sign indicating whether it lies inside or outside the structure. This transformation converts the more difficult mesh-to-volume registration problem into a well-established volume-to-volume registration framework (Balakrishnan et al., 2019; Hu et al., 2018; Chen et al., 2022; Guo et al., 2024; Liu et al., 2024). Our approach leverages a recently developed vector field attention network (VFA) (Liu et al., 2024) to register SDM to the target image, thereby transforming the canonical mesh to align with the anatomical structures to be segmented. Before registration, the canonical mesh is roughly aligned with the target structure using a single-center prompt provided by the user. This reduces the need for multiple point prompts required by SAM to resolve ambiguity. The registration-based approach also provides additional benefits, such as point-to-point correspondence, compared to techniques that directly predict binary masks. While previous studies, such as (Lee et al., 2019), have

explored similar concepts of registering a fixed structural template to an image, our goal is to adapt the registration behavior based on the prompted structure thus paving the way for more generalizable models. We achieve this by investigating new mesh representations, loss functions, and more advanced registration networks.

Compared to segmentation via direct image registration, our method has two key advantages. First, individual structures are registered independently, allowing for flexible initialization and deformation. Second, we compute losses based on the warped mesh boundaries rather than region-based metrics such as mean squared error or DICE, enabling higher accuracy in point correspondence on anatomical surfaces. Experiments demonstrate that our approach achieves performance similar to image registration in terms of region overlap but significantly outperforms it in point correspondence accuracy. This is particularly beneficial for applications requiring precise point localization or organ parcellation, such as distinguishing heart chambers or vascular structures.

We evaluate our approach using two datasets: (1) head CT images containing ear structures, where we found that MedSAM faces challenges, and (2) body CT images where MedSAM has been successfully applied (Ma et al., 2024; Zhu et al., 2024; Lei et al., 2024).

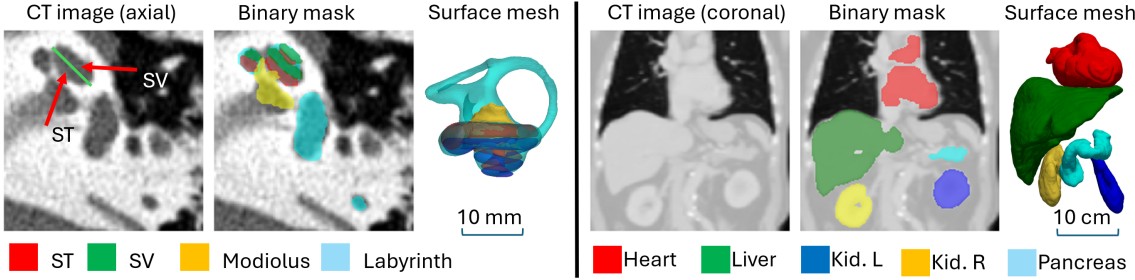

Figure 1: Illustration of the cochlear (left) and body (right) datasets.

## 2. Methodology

Figure 2 provides an overview of our proposed mesh-prompted segmentation framework. Given a test image f

### 2.1. Vector Field Attention Network (VFAN)

Let the fixed (target) and moving (source) images be $I_f$ and $I_m$, respectively, VFAN warps $I_m$ to match $I_f$ by predicting a transformation $\phi$. This transformation maps each location $x$ in $I_f$ to a corresponding location in $I_m$, and the intensity of the warped image $I_w$ is obtained by sampling $I_m$ at $\phi(x)$. To predict $\phi(x)$, VFAN first uses a feature extractor network to generate feature maps $F$ for $I_f$ and $M$ for $I_m$ at multiple resolutions. At each resolution $i$, a neighborhood $N(x)$, defined as a $3 \times 3 \times 3$ grid centered at $x$, is constructed. The feature vector of $I_f$ at location $x$, denoted $F^i(x)$, is compared with the feature vectors of $M$ at every location $y \in N(x)$, producing a $3 \times 3 \times 3$ similarity map. This similarity map is normalized using Softmax to create an attention map $A(x)$ at $x$. Let $A_y(x)$ denote the attention value of $x$ for a neighbor $y$, the transformation $\phi(x)$ is computed using the following equation:

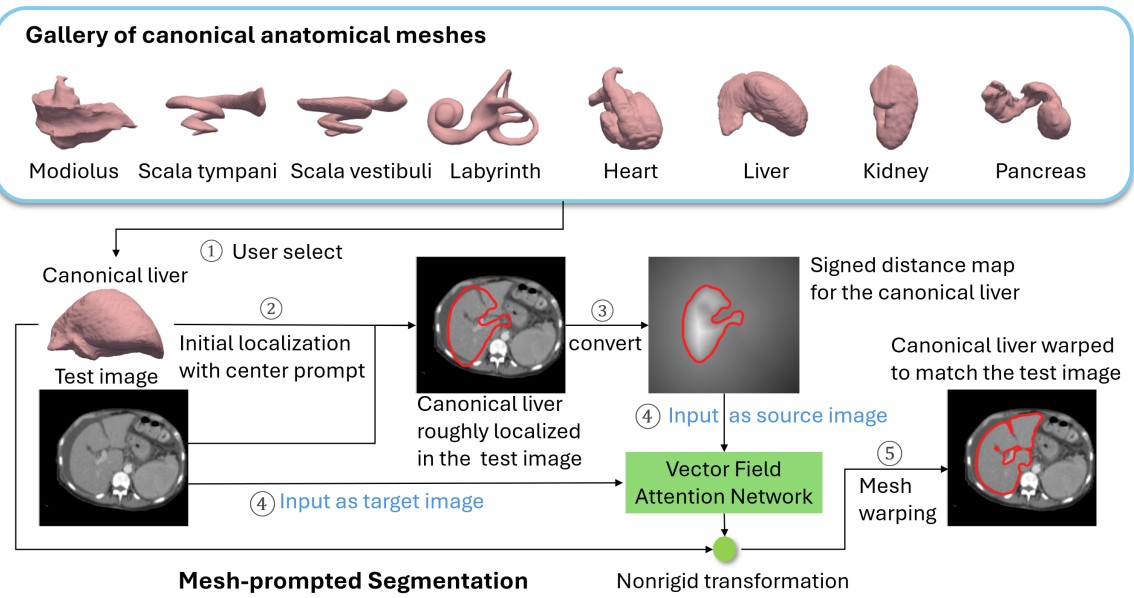

Figure 2: Workflow of mesh-prompted segmentation.

$$\phi(x) = \sum_{y \in N(x)} A_y(x)(y - x) \tag{1}$$

The above equation converts the similarity in feature space to a displacement vector. These local displacements are combined across different resolutions to capture long-range correspondences effectively.

## 2.2. Training VFAN for Mesh Prompted Segmentation

To implement MPS with VFAN, which we call MPS-Reg, we integrate structures represented by both binary masks and meshes. Let $M_m$ and $B_m$ denote the mesh and binary mask for the structure in the source space, and $M_f$ and $B_f$ denote those for the structure in the target space. The goal is to transform $M_m$ into $M_w$ and $B_m$ into $B_w$ such that they best match $M_f$ and $B_f$, respectively. This transformation can be achieved using one of two approaches. In the first approach, referred to as "pulling", target coordinates are transformed into the source space, $B_w$ is generated by sampling from $B_m$, and $M_w$ is obtained from $B_w$ using the marching cubes algorithm. Alternatively, in the second approach, referred to as "pushing", the transformation is performed in the opposite direction (source to target). In this case, $M_m$ is directly transformed into $M_w$, and $B_w$ is generated from $M_w$ using a ray-casting algorithm (Shimrat, 1962). For our proposed MPS, we adopt the pushing scheme, as it offers higher accuracy for localizing mesh points in the test image, as shown in the results section. Specifically, we utilize VFAN, which takes the test image and the SDM of the prompted mesh as inputs and predicts the transformation that maps the mesh points to the test image space. Inspired by (Lin et al., 2024), we use a hyperbolic form of Chamfer

Distance as the loss function to compare $M_w$ and $M_f$ in the test image space. The hyperbolic Chamfer Distance between two sets of vertices $X$ and $Y$ is defined as:

$$D(X,Y) = \frac{1}{|X|} \sum_i \min_j d(X_i, X_j) + \frac{1}{|Y|} \sum_j \min_i d(X_i, X_j) \tag{2}$$

where $d(X_i, Y_j)$ is the hyperbolic distance between the $i^{th}$ point in $X$ and $j^{th}$ point in $Y$ which is defined as,

$$d(X_i, Y_j) = arcosh(1 + 2\frac{||X_i - Y_j||^2}{(1 - ||X_i||^2)(1 - ||X_j||^2)}) \tag{3}$$

The hyperbolic Chamfer Distance is more robust to outlier points because it reduces the penalty for point pairs with large distances. This robustness is crucial, as the same structure from different subjects may include small regions that lack direct correspondence. By using this distance metric, the focus is placed on the overall shape rather than minor irregularities. During each training iteration, we randomly select an image from one subject and a structure mesh from another subject in the training set. The mesh is then positioned around the ground-truth center with a random shift ranging from 0 to 5 voxels in each of the x, y, and z directions. The image and the derived SDM are then processed through VFAN to compute $D(M_w, M_f)$, and backpropagation is performed to update the model.

## 2.3. Dataset

We used two datasets in our experiments:

(1) Head CT images: This dataset includes structures of the modiolus, scala tympani, scala vestibuli, and the labyrinth (Wang et al., 2021). A total of 367 images are used for training, 50 for validation and 87 for testing. Ground-truth meshes are generated using active shape models (Noble et al., 2011). Because of the use of active shape models, we know the ground-truth point-to-point correspondences between meshes from different patients. This allows us to assess registration performance using point-to-point error, which we will present in Section 3.2. All images are resampled to have an isotropic voxel size of 0.2 mm and are cropped to a size of $128 \times 128 \times 128$.

(2) Body CT images: These images are used for training Totalsegmentor (Wasserthal et al., 2023). From a set of 1230 images, we select a subset of 285 images that fully contain the heart, liver, bilateral kidneys and pancreas. Of these, 205 images are used for training, 30 for validation and 50 for testing. Ground-truth meshes are generated from binary masks, which are obtained through automatic methods and manually refined, using the marching cubes algorithm. However, since the marching cubes algorithm does not preserve point-to-point correspondences across subjects, we use Chamfer distance rather than point-to-point error to evaluate the surface registration error on this dataset. All images are resized to $128 \times 128 \times 128$ crops, with voxel sizes ranging from $2.3 \times 1.7 \times 2.1$ to $4.0 \times 4.0 \times 4.5$ mm.

We train separate models for each dataset. For both datasets, one subject is chosen as the atlas to provide the canonical meshes used for inference. 2. For the cochlear dataset, the atlas is chosen to have average anatomical sizes, as measured by two axis lengths of the ST structure commonly used in the literature (Salamah et al., 2023). For the body dataset, there was no clear rule for selecting canonical prompts, so a random atlas was used.

## 2.4. Baseline methods

We compare our method with three alternative approaches for anatomy segmentation:

1. nnU-Net (Isensee et al., 2020): This method trains a U-Net from scratch in a fully supervised way using DICE loss. We train nnU-Net on each dataset for 200 epochs using a full-resolution 3D model with 1-fold cross-validation. For the cochlear dataset, we train separate models for each structure to address the overlapping issue.

2. MedSAM2 (Zhu et al., 2024): This method builds on SAM2 (Ravi et al., 2024) and segments 3D medical images as videos consisting of 2D slices using points prompted on one or many slices. Following the approach outlined in (Zhu et al., 2024), we load pretrained weights from SAM2 and fine-tune the model on each dataset for 200 epochs (no performance improvement was observed with additional training). For the body dataset, we use the ground-truth bounding box to prompt the pretrained model for every two axial slices. For the cochlear dataset, since defining accurate bounding boxes for each structure is not feasible due to intertwined shape, we instead use one randomly selected interior point for every two axial slices as prompts. If two structures overlap, we select points from their exclusive region.

3. Image-Reg: This approach performs segmentation by registering an atlas image to the test images using VFAN. The model is trained in the "pulling" scheme with DICE loss used to compare the warped and ground-truth binary masks and a gradient-based regularization term employed to improve the smoothness of the transformation.

## 3. Results

### 3.1. Segmentation Accuracy in terms of Region Overlap

We compare segmentation accuracy across different methods using the DICE score on the test set. As shown in Fig. 3A. MedSAM2 produces inferior results compared to MPS-Reg. As expected, it struggles to preserve the shape of cochlear structures due to ambiguity with point prompts. For example, it generates incomplete labyrinths (red circle) and confuses ST and SV (yellow circle), resulting in very low DICE scores on the cochlear dataset. While it performs better on the body dataset, the segmented structures still lack realism, e.g., vessels connected to the heart are often merged (black circle), making it less accurate than MPS-Reg. nnU-Net performs well on the cochlear dataset but poorly on the body dataset, which has greater variation in organ shape and position. This leads to overfitting given the relatively small training set: predictions are accurate only for images similar to the training set. As a result, the DICE scores for nnU-Net on the body dataset show a much higher standard deviation compared to those on the cochlear dataset. The differences observed between MedSAM2 and nnU-Net align with the findings reported in (Zhu et al., 2024). The slightly lower performance of MedSAM2 in this work may be attributed to differences in the dataset and the relatively low image resolution used.

| | | Cochlear | | | | | Body | | | | | |
|---|---|---|---|---|---|---|---|---|---|---|---|---|
| Model | | MD | ST | SV | Laby. | Avg | Heart | Liver | Kid. L | Kid. R | Pancreas | Avg |
| nnU-Net | | 0.861 | **0.909** | **0.899** | **0.931** | **0.900** | 0.900 | 0.943 | 0.773 | 0.819 | 0.542 | 0.795 |
| | ± | 0.102 | **0.047** | **0.049** | **0.024** | **0.055** | 0.178 | 0.072 | 0.349 | 0.322 | 0.274 | 0.239 |
| MedSAM2-FT | | 0.630 | 0.662 | 0.624 | 0.820 | 0.684 | 0.836 | 0.888 | 0.881 | 0.913 | 0.607 | 0.825 |
| (FT: Fine-Tuned) | ± | 0.063 | 0.044 | 0.053 | 0.052 | 0.053 | 0.043 | 0.043 | 0.090 | 0.039 | 0.147 | 0.073 |
| Image-Reg | | **0.906** | 0.891 | 0.888 | 0.897 | 0.895 | *0.938* | *0.957* | *0.905* | **0.940** | *0.788* | *0.905* |
| | ± | **0.033** | 0.039 | 0.041 | 0.019 | 0.033 | *0.021* | *0.014* | *0.149* | **0.071** | *0.113* | *0.074* |
| MPS-Reg | | *0.872* | *0.903* | *0.891* | *0.923* | *0.897* | **0.944** | **0.964** | **0.912** | *0.936* | **0.807** | **0.913** |
| | ± | *0.039* | *0.046* | *0.049* | *0.022* | *0.039* | **0.014** | **0.009** | **0.104** | *0.027* | **0.095** | **0.050** |

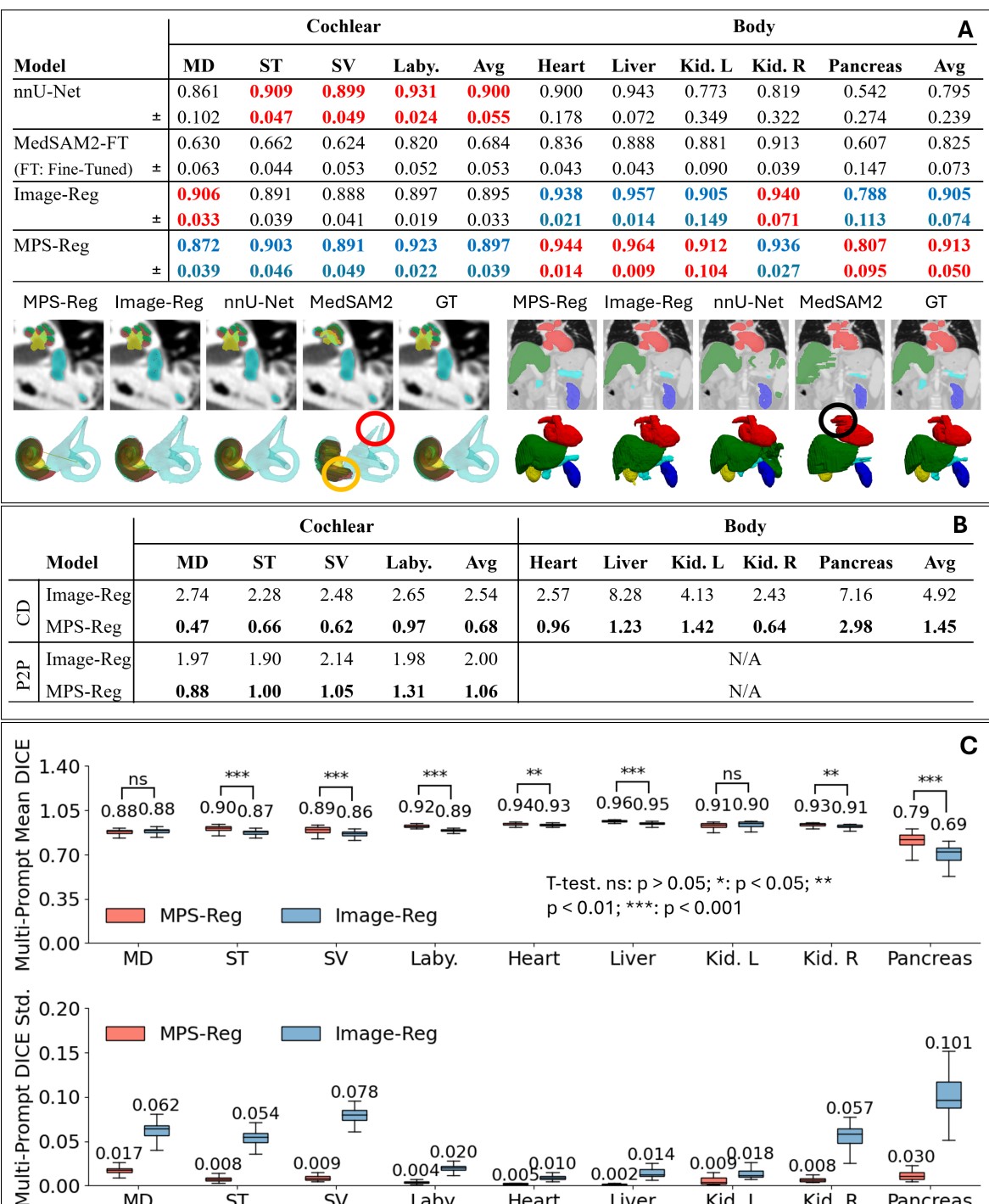

| | | Cochlear | | | | | Body | | | | | |
|---|---|---|---|---|---|---|---|---|---|---|---|---|
| | Model | MD | ST | SV | Laby. | Avg | Heart | Liver | Kid. L | Kid. R | Pancreas | Avg |
| CD | Image-Reg | 2.74 | 2.28 | 2.48 | 2.65 | 2.54 | 2.57 | 8.28 | 4.13 | 2.43 | 7.16 | 4.92 |
| CD | MPS-Reg | **0.47** | **0.66** | **0.62** | **0.97** | **0.68** | **0.96** | **1.23** | **1.42** | **0.64** | **2.98** | **1.45** |
| P2P | Image-Reg | 1.97 | 1.90 | 2.14 | 1.98 | 2.00 | | | N/A | | | |
| P2P | MPS-Reg | **0.88** | **1.00** | **1.05** | **1.31** | **1.06** | | | N/A | | | |

Figure 3: A: Segmentation performance evaluated by the mean and standard deviation of the DICE across test cases, accompanied by visual examples of an average case. Red and blue texts show the best and second best results in each column. B: Segmentation performance evaluated by surface error. C: Mean and variance of DICE with 10 different prompts. Boxes are plotted across test cases.

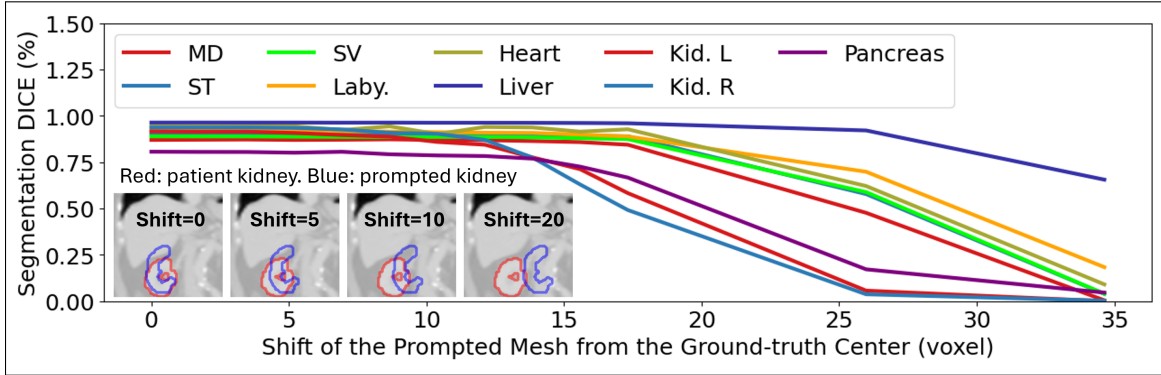

Figure 4: Performance of MPS-Reg with inaccurate initial localization. Visual examples of shifts in the prompted kidney are provided for better interpretation of the shift.

## 3.2. Segmentation Accuracy in terms of Surface Error

In Fig. 3B, we compare the surface error for each structure after registration using MPS-Reg and Image-Reg. For both methods, surface error is measured by Chamfer distance (CD) and point-to-point correspondence error (P2P), calculated after transforming the atlas mesh into the patient space. This transformation can be useful as it transfers atlas information to the patient. Note that we do not compare surface error with nnU-Net or MedSAM2 because these methods segment structures by predicting binary masks rather than transforming mesh points, making DICE a more suitable evaluation metric. Additionally, computing P2P or CD for these methods would require registering the atlas mesh to the mesh reconstructed from the binary mask using a non-rigid point cloud registration technique. This introduces irrelevant errors and makes the comparison less informative.

Fig. 3B shows that MPS-Reg achieves remarkably lower surface error than Image-Reg. The superiority in Chamfer distance is expected, as MPS-Reg is trained with a hyperbolic Chamfer distance loss. What is more notable is that, despite not explicitly enforcing point-to-point correspondence during training, MPS-Reg achieves excellent P2P accuracy (P2P error can only be evaluated on the cochlear dataset for which the ground-truth meshes are generated by active shape models, which naturally preserves point correspondence across subjects). Therefore, for applications requiring precise surface localization in patients, MPS-Reg provides a more reliable solution.

## 3.3. Analysis of Model Robustness

This section examines factors affecting segmentation performance. First, we fix the initialization of each structure at the ground-truth center and use meshes from different subjects as prompts to analyze how DICE scores vary with different prompts. Then, we keep the prompted mesh fixed and evaluate segmentation accuracy while varying the initial location.

In Fig. 3C, we show the mean and variance of DICE across 10 trials of MPS-Reg for each structure, using meshes from different subjects in the validation set as prompts. For comparison, we also report the results for Image-Reg, with 10 trials using images from different subjects. The low multi-prompt DICE variance of MPS-Reg indicates that model

performance is largely independent of the choice of the prompted mesh when properly initialized. In contrast, Image-Reg exhibits a much higher variance (and lower mean DICE) across multiple prompts, as it segments all structures simultaneously. Therefore, differences between the atlas and test image are influenced not only by individual structure shapes but also by the relative positions of structures, making the choice of atlas images more impactful. This highlights a key advantage of segmenting structures individually with MPS-Reg over joint segmentation with Image-Reg.

In Fig. 4, we analyze how shifts in the initial position from the ground-truth center affect MPS-Reg's segmentation performance. The model remains stable with shifts up to 10 voxels (in random directions). Beyond this, performance begins to degrade, with smaller structures such as the kidneys and pancreas declining more rapidly. The segmentation of larger structures, like the liver, remains accurate until shifts reach 25 voxels. This behavior aligns with intuition, suggesting that the initial overlap between the structure in the test image and the SDM is crucial. Note that this relationship is influenced by the random shifts applied during training, an important hyperparameter that warrants further investigation.

## 4. Discussion

This paper introduces a novel approach for anatomy segmentation using mesh prompts in image-to-image registration. Compared to point prompts commonly used in contemporary vision foundation models, mesh prompts significantly reduce ambiguity and better preserve the shape of the structures. Compared to conventional image registration, our method demonstrates greater robustness to the choice of atlas and achieves improved point correspondence on the structure's surface, although it requires slightly more effort from the user to select a mesh and specify the center. Importantly, our approach introduces a potential method for a new type of foundation model capable of segmenting previously unseen structures using mesh prompts after large-scale training. This concept forms the core motivation behind proposing MPS, which we intend to investigate further in future work.

The current work still has a few limitations. First, the MPS framework assumes anatomical structures have predictable shapes, which may make it less effective for pathological structures or tumors. Second, whether SDM is the optimal mesh representation has not been fully explored. Using learnable representations, such as those produced by PointNet (Qi et al., 2017), could potentially improve registration, though this would require innovations in the registration network. Third, while we have justified the choice of VFAN for registration and hyperbolic Chamfer distance as the loss function, a rigorous ablation study comparing alternative combinations has yet to be conducted.

### Acknowledgment

This work has been supported by National Institute of Health (NIH) grants R01DC014037, R01DC014462 and R01DC008408 and by the Advanced Computing Center for Research and Education (ACCRE) of Vanderbilt University. This work is also supported by National Cancer Institute (NCI) grants R01CA253923 and R01CA275015. The content is solely the responsibility of the authors and does not necessarily represent the official views of these institutes.

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
