# OpenReview forum: "Mesh-Prompted Anatomy Segmentation"
_MIDL.io/2025/Conference — MIDL 2025 Oral_

### Official Review · Reviewer_WiAj · 2025-02-20

**Confidence:** 5
**Preliminary Rating:** 5
**Recommendation:** Oral, Poster

**Summary:**

This paper presents a template-based method for medical image segmentation.
The template is roughly placed at the correct location with a user click, and then gets deformed to fit the target structures.
The deformation is generated by a neural network from the signed distance map of the template and the input image.
The method is evaluated on two clinical applications, namely cochlear and abdominal organs segmentation. The results show that the algorithm produces segmentations that are comparable to standard approaches like nnUnet in terms of Dice but with a low point distance.

**Strengths:**

* Since this method relies on a template deformation, it provides segmentation results with low point-to-point distances and can potentially be used for downstream tasks like landmark detection
* The method is very clearly described and paper is nicely written in general
* The experiments are based on two relatively large public datasets. They include various statistics like Dice but also point-to-point distances, as well as the stability w.r.t. the template and robustness to initial positioning.

**Weaknesses:**

* Neither the code not the models seem publicly available
* I find the title misleading - one would expect from "mesh prompted segmentation" a similar encoding of the mesh than clicks, bounding boxes or masks from SAM2. It is not wrong, but it seems to introduce a new major methodology whereas deforming a segmentation mask (or a distance field) has been done for years (see next point).
* Following on this point, some related work has not been discussed:
[Real-time 3D image segmentation by user-constrained template deformation](https://link.springer.com/chapter/10.1007/978-3-642-33415-3_69) deforms a signed distance map for segmentation purposes
[TETRIS: Template transformer networks for image segmentation with shape priors](https://ieeexplore.ieee.org/abstract/document/8672808/) learns to deform a segmentation template for segmentation purposes
* A single number to quantify the performance of a method on a dataset is not enough. There should be either boxplots, or at least standard deviations. Furthermore, statistical tests have to be performed to try to estimate whether some differences are significant (e.g. 0.909 vs 0.903).

**Detailed Comments:**

* The challenge with shape-based methods is often to be able to capture abnormalities/pathologies - robustness often come with a reduced flexibility. It would have been nice to see some results on the flexibility of the method, or at least a qualitative discussion.
* How was the canonical mesh chosen - Just randomly? Shouldn't it be some kind of mean template? Would it be possible to make it trainable as well, so that the method could automatically find the "best template"?
* The point to point distances are only compared between MPS and Image-Reg. I understand why - but it would have been possible, as another baseline, to register the template to the segmentation produced by nnUnet and to estimate the point the point distances.
* I am not sure to understand the benefits of the (relatively complex) hyperbolic Chamfer Distance with the arcosh. The authors mention the reduced penalty for outliers, but have you tried other more standard losses like Huber for instance?
* I am a bit surprised by the visual results of nnUnet in Figure 3A, in particular the obvious wrong liver detections in the other organs. nnUnet is considered one of the most challenge-winning approach for segmentation. Has it been trained for too few epochs? Do you have an explanation?
* What does MedSAM2-FT means ? (the FT)

**Justification Of The Preliminary Rating:**

This paper is overall well written and easy to follow.

While there are definitely some things that could be improved for a final version, none of them are major flaws so I am in favor of its acceptation. I still hope the authors will answer some of the questions I have raised.

**Questions To Address In The Rebuttal:**

* Are you going to make the code available?
* Why did you choose MedSAM2 as a baseline? A 2D approach does not seem like the best method for the task at hand (which actually shows in the performance comparison)
* Can you discuss the relationship to [TETRIS: Template transformer networks for image segmentation with shape priors](https://ieeexplore.ieee.org/abstract/document/8672808/) ? Does using a signed distance transform rather than a binary mask have a big impact?
* The advantage of using a template, especially with complex shapes like the cochlear, is to have some topology constraints, but this method does not guarantee that the deformation stays diffeomorphic, does it? Have you noticed topological issues in your predictions?
* see comments above

**Special Issue:**

Yes

---

> ### Author Response · Authors · 2025-03-07
> **Rebuttal Response - Part 1**
>
> We thank Reviewer 3's detailed comments and address their questions as follows:
>
> 1.	Yes, we will make the code available. We are currently cleaning up the code and will put the link in the final manuscript if accepted.
>
> 2.	We agree that 2D approaches may be suboptimal due to their lack of spatial consistency. This actually is the reason for our choice of MedSAM2 over other SAM-based foundation models such as MedSAM. MedSAM2 attempts to capture spatial relationships across 2D slices by treating the 3D volume as a sequence of 2D frames, making it an approach between purely 2D and fully 3D ones. More importantly, MedSAM2 has been reported to outperform many 3D methods, including nnUNet and SAM-Med3D, on a body CT dataset similar to ours.
>
> 3.	Thank you for bringing the paper TETRIS: Template Transformer Networks for Image Segmentation with Shape Priors to our attention. We have cited it in the revised manuscript and discussed its differences from our approach. Our primary contribution is demonstrating the possibility of changing the registration behavior based on the prompted mesh (e.g., registering a liver and a heart can be very different), which is a first step towards building foundation segmentation models. Regarding the choice between binary masks and SDMs, we favor SDMs because binary masks result in 3×3×3 convolution kernels encountering large uniform regions that suddenly change at boundaries, potentially affecting network gradient properties. However, whether SDM is the optimal representation has not been fully studied, and we have acknowledged this as a limitation in the Discussion section.
>
> 4.	The topological issue is an excellent question. We did not assess this property in the current work. However, in our previous study on VFNA architecture (citation provided below), extensive analysis showed that the percentage of non-diffeomorphic voxels after warping with VFNA is around 0.06%, significantly lower than commonly used methods such as VoxelMorph and TransMorph. The non-diffeomorphic rate can be further reduced to below 0.01% using a scaling-and-squaring technique introduced in that study. In this work, we did not observe major topological issues, so we did not focus on this aspect.
>
> Yihao Liu, Junyu Chen, Lianrui Zuo, Aaron Carass, and Jerry L. Prince. Vector field attention for deformable image registration, 2024.

---

> ### Author Response · Authors · 2025-03-07
> **Rebuttal Response - Part 2**
>
> For the additional questions in the Detailed Comments and Weaknesses:
>
> 1.	We acknowledge the trade-off between robustness and flexibility. Our core assumption is that normal human organs follow predictable shapes, which implies potential performance degradation for abnormal structures. This limitation is now discussed in the revised manuscript.
>
> 2.	For the cochlear dataset, canonical prompts were selected based on average sizes, as measured by two axis lengths of the ST structures established in the literature. For the body dataset, there was no clear rule for selecting canonical prompts, so a random selection was used. Figure 3C illustrates that model performance is not highly sensitive to template choice, though automatically learning or selecting the template remains an interesting direction for future work.
>
> 3.	Yes, it is possible to reconstruct a mesh from the binary mask, register the atlas mesh to it using non-rigid point cloud registration, and then compute P2P and CD. We explored this approach in a related study but found that the registration process itself introduced complexity and errors, making the comparison less informative. Therefore, in this work, we limit P2P and CD evaluations to methods that directly transform points from the atlas to the patient. We have explained this in section 3.2 in the revised manuscript.
>
> 4.	The choice of hyperbolic Chamfer Distance was motivated by the need to reduce the influence of irregular points, which are more common in the body dataset, where meshes are derived from binary masks. Additionally, previous studies have shown that hyperbolic Chamfer Distance improves point cloud reconstruction compared to alternative metrics. Further ablation studies are necessary to fully explore this choice, but time did not permit to conduct them and report the results in the current work.
>
> 5.	We attribute the poor performance with nnU-Net on the body dataset to the high variability in body CT images and the relatively small training set - the shape of body structures show much higher variance than the cochlear structures but it has even less training data. In our experiments, we observed that nnUNet exhibited large variations in Dice scores for the body dataset, likely due to overfitting—performing well on cases similar to the training set but poorly on others. This point is discussed in our result section and has been made clearer in the revision after adding the standard deviation of the DICE in Figure 3.A.
>
> 6.	In MedSAM2-FT, "FT" stands for fine-tuning. MedSAM2 was fine-tuned on the two datasets used in this study. We have added this clarification in the table.
>
> 7.	We have included the standard deviation of the Dice scores in Figure 3A. The suggested significance tests are not provided for Figure 3A but we have added some relevant tests in Figure 3C.
>
> 8.	After careful consideration, we have revised the title to Shape-Guided Anatomy Segmentation Using Mesh Prompts to better reflect the segmentation process and highlight the role of mesh prompts.

---

### Official Review · Reviewer_T1jr · 2025-02-21

**Confidence:** 4
**Preliminary Rating:** 5
**Final Rating:** 5

**Summary:**

The manuscript presents a a mesh prompt generation algorithm for SAM-based segmentation of cochlear and thoracic-abdominal structures. It proceeds by converting reference meshes to signed distance functions subsequently registering the resulting image to the test image using a learning-based registration algorithm which uses a metric robust to outliers. The approach is evaluated on two datasets and compared favorably to 3 SOTA methods, including nnU-Net and MedSAM2, demonstrating higher, more stable performances, and the poor performances of point-based prompts in particular.

**Strengths:**

The use of pretrained SAM models for downstream segmentation tasks with little training, mainly to align reference meshes with test images to create prompts. Compared to point-based prompts, the method would require little to no further interaction from the operator or the prompt generation algorithm.

**Weaknesses:**

I cannot spot any. Given the high technicality of the paper, it is very well written, with sufficient level of detail and quite thorough performance evaluation and presentation of quantitative and qualitative results.

**Detailed Comments:**

The manuscript doesn't give any execution time measurements, in particular to assess the speed of the registration method required to align the mesh prompt with the target image. In addition, authors ought to cite their previous work which they mention at the end of Section 2, and  clarify the differences therein with the present manuscript.

**Justification Of The Final Rating:**

Following the review and discussion phases, I think the improvements and clarifications authors brought have bettered an already good manuscript which deserves the attention of the specialist public that takes interest in the use of pretrained SAM models for downstream segmentation tasks with little training via robust automatically-generated prompts.

**Justification Of The Preliminary Rating:**

The manuscript is sound both on methodology and form, it gives insights allowing to understand the shortcomings of recent SAM models and proposes an important improvement via mesh prompt generation, which could be easily applied to the segmentation of other organs and pathologies.

**Questions To Address In The Rebuttal:**

Please address the points I have mentioned in the detailed comments section.

---

> ### Author Response · Authors · 2025-03-07
> **Rebuttal Response**
>
> We thank Reviewer T1jr's comments and address their questions as follows:
>
> 1.	As is also noted by Reviewer 1, computing SDM indeed can be computationally expensive. To mitigate this, we precompute the SDM for all training structures offline before training begins. During inference, we also precompute the SDM for all prompted structures, ensuring that SDMs do not introduce additional computational overhead. As a result, the time needed to align the mesh prompts is similar to aligning two images using standard deep-learning-based registration techniques such as Voxelmorph.
>
> 2.	We have added the missing citation.

---

> > ### Comment · Reviewer_T1jr · 2025-03-11
> >
> > I thank the authors for the improvements made to the manuscript. Unfortunately they don't reply adequately to my question related to the speed of the registration method for which I expect a quantitative measure, put in perspective with respect to the total processing time in training and inference. I duly note that precomputing the SDM for prompted shapes at inference time would limit the user to a library of predefined structures.

---

> > > ### Author Response · Authors · 2025-03-13
> > >
> > > Thank you for clarifying your questions. To provide a more precise response, we quantitatively analyzed the time required for both training and inference.
> > >
> > > On an RTX 2080Ti GPU, the model’s forward pass for registering two 128×128×128 volumes takes 0.14 seconds (7.54 seconds on a 2.2GHz CPU). Each training iteration (including data loading, forward pass, and backpropagation) with a batch size of 1 takes 1.68 seconds. In our experiments, we train the model for 100000 iterations, which takes around 46 hours. For inference, segmenting a single structure on one volume (including data loading and forward pass) takes 1.24 seconds.
> > >
> > > SDM computation is performed during preprocessing using scipy.distance_transform_edt, which takes an average of 0.91 seconds per structure. Regarding the concern about SDM precomputation, users can still employ their own SDMs. For example, they can compute the SDM for one structure, store it, and use it to segment 100 volumes, which will take 0.91 + (1.24 × 100) = 124.91 seconds. If they do not store the SDM and compute it every time, which is unnessary if the prompt is not changed, the time needed is (0.91 + 1.24) x 100 = 215 seconds.
> > >
> > > We will add the time measurements in the final manuscript and provide all related codes, if this paper gets accepted.

---

### Official Review · Reviewer_3gtC · 2025-02-21

**Confidence:** 4
**Preliminary Rating:** 3
**Final Rating:** 4

**Summary:**

This paper introduces a novel "Mesh-Prompted Segmentation" method for anatomical structure segmentation in medical images. The authors argue that mesh-based prompts are better than point-based prompts. They use a canonical or template mesh as a prompt, which acts as a stronger shape prior to overcoming ambiguities in the structure of the organ of interest. The method registers the signed distance function of the template mesh to the target image using a vector-field attention network (VFAN). A single center point from the user roughly aligns the mesh in the target image coordinate system. The VFAN then learns to deform the mesh to fit the target anatomy.

**Strengths:**

1. Using a canonical mesh provides a powerful shape prior, enhancing robustness in challenging medical images with noise, blurriness, and overlapping structures.
2. The method allows independent registration of individual structures, enabling flexible initialization and deformation.
3. Using a hyperbolic Chamfer distance enables higher accuracy.

**Weaknesses:**

1. The overall flow of the paper could be improved as it would make the paper easier to read. The methods section is especially unclear, making it difficult to understand the proposed model and leaving many questions unanswered.

2. Although the authors call it the mesh-prompted method, technically, the only prompt from the user is the center of the anatomy of interest. The mesh used is a canonical template mesh (which already exists). So, maybe the model's name and motivation could be restructured to convey that. Calling it mesh-guided anatomy segmentation might be more appropriate.

**Detailed Comments:**

1. The methods section could benefit from a rehaul. The block diagram of the model is not very intuitive and does not provide a clear picture of the flow of the method. The authors should consider specifically showing and explaining the following steps (if the understanding is correct here):
(a) Select target mesh (Mf) and target Image, (b) Deform Mf to source mesh Mm and store the transformation. (c) Given that transformation and the target image, reconstruct Mf from Mm (Mf'). (d) Calculate Chamfer distance between Mf and Mf'.
2. The paper could benefit from more ablation studies to analyze the contribution of different components of the method (e.g., the VFAN architecture and the hyperbolic Chamfer distance).
3. Figure 3.C is confusing, and the numbers don't match the Table in Figure 3.A. If you perform multiple trials of the experiment, it would be more logical to show a box plot of the DICE scores, which conveys the mean DICE score and the variation across multiple trials. Fig 3.C only conveys the variation, which is meaningless on its own.

**Justification Of The Final Rating:**

The authors have provided detailed responses to my concerns and have made meaningful revisions to clarify their methodology.
However, some concerns—such as additional ablation studies and a deeper evaluation of correspondences—remain open.

**Justification Of The Preliminary Rating:**

There are some concerns about calling this a mesh-prompted segmentation model and the methodology and figures require significant revisions to convey the idea clearly. Moreover, unclear presentation of DICE scores limits the justification of superiority in performance.

**Questions To Address In The Rebuttal:**

1. The authors mentioned that the proposed method comes with the added advantage of establishing correspondences across the samples. Have they tried to test the quality of these correspondences by performing statistical analysis on the segmented anatomies? Look at the PCA modes of variation to analyze population statistics.
2. Registering the SDM might be computationally expensive, especially for high-resolution images and complex meshes. The paper does not provide detailed information on computational time.
3. Why do you need mesh-based input here if you are converting it into an SDM anyway?
4. Why have the authors not reported the CD and P2P for other methods in Figure 3.B, and what is the reason for the lack of P2P metrics for the Body dataset?
Also, check the detailed comments section.

---

> ### Author Response · Authors · 2025-03-07
> **Rebuttal Response**
>
> We thank Reviewer 3gtC's detailed comments and address their questions as follows:
>
> 1.	For the cochlear dataset, ground-truth segmentations were obtained using active shape models, which are similar to statistical point models based on eigenmodes, as you suggested. These models naturally provide silver-standard ground-truth correspondences, which we use to compute the P2P error reported in Figure 3B. In contrast, the body dataset provides ground-truth segmentations as binary masks without point-to-point correspondences, making it impossible to compute the P2P error for this dataset. We have clarified this in the revised manuscript.
>
> 2.	Computing SDM is indeed computationally expensive. To mitigate this, we precompute the SDM for all training structures offline before training begins. During inference, we also precompute the SDM for all prompted structures, ensuring that SDMs do not introduce additional computational overhead.
>
> 3.	Our technique incorporates mesh representations in two ways: first, by converting the mesh into an SDM as input to the network, and second, by computing the Hyper-Chamfer loss using points on the mesh. Because of this dual dependence on mesh representations, we classify our approach as mesh-based rather than purely SDM-based.
>
> 4.	Computing P2P error with nnU-Net or MedSAM2 is not straightforward. Both methods generate segmentations as binary masks rather than transforming points from an atlas to patient space, which disrupts direct point correspondences. In theory, if we have ground-truth P2P correspondence, as in the cochlear dataset, P2P error could be estimated by reconstructing a mesh from the binary mask, registering the atlas mesh to the reconstructed one using non-rigid point cloud registration, and then comparing the transformed mesh to the ground-truth mesh. However, point cloud registration introduces additional errors, making this comparison less informative. While computing Chamfer distance (CD) with nnU-Net and MedSAM2 might be simpler than computing P2P, we do not think it would provide additional insights, as we have already compared DICE. Therefore, for simplicity and clarity, we only compare CD and P2P for MPS-Reg and Image-Reg, which directly transform the mesh points. This explanation was missing in the original manuscript and has been clarified in the revision.
>
> For the additional questions in the Detailed Comments and Weaknesses:
>
> 1.	Your understanding of our technique’s workflow is correct, and we have incorporated your suggestions into the Methods section to clarify the processing steps.
>
> 2.	We agree that an ablation study on network structure and loss functions would enhance the comprehensiveness of our study. We actually have explored different alternatives before settling down on VFAN and hyperbolic Chamfer distance, but a rigorous study requires more efforts, and time did not permit to fully conduct the experiments. We have acknowledged this as a limitation in the Discussion section.
>
> 3.	We agree that reporting the mean Dice across trials provides useful information, and we have added this information in Figure 3C. The mean Dice values in Figure 3C are not expected to match those in Figure 3A, as the latter represents results from a single trial, whereas Figure 3C presents the mean Dice across multiple trials. The prompted meshes in our methodology are intended to be canonical representatives. For the cochlear dataset, they were selected based on average sizes, while for the body dataset, we lacked a clear rule for selecting canonical prompts and therefore used a randomly chosen one. Given this, both Figure 3A (evaluating performance with one special prompt) and Figure 3C (using different prompts) provide useful insights into our technique’s performance.
>
> 4.	Title. We have carefully considered the comments on the title from both Reviewer 1 and 3 and decided to change the title to be: Shape-guided Anatomy Segmentation using Mesh-Prompts. We still use the word “prompt” because despite the fact that the meshes have already been predefined, the user still needs to select which mesh to use from the gallery, which reflects the user intention to some extent.

---

> > ### Comment · Reviewer_3gtC · 2025-03-14
> >
> > I thank the authors for considering the change in the paper's title. It reflects the proposed method better now.
> > Also, the change in the figures is appreciated.

---

### Author Rebuttal · Authors · 2025-03-07

**Rebuttal:**

We appreciate all reviewers for their positive feedback and constructive comments. To further improve the quality of this work based on their suggestions, we have made the following revisions to the manuscript:

1. Modified the title to better reflect the core focus of this study.
2. Added standard deviation values for DICE in Fig. 3A.
3. Included box plots of mean multi-prompt DICE with significance tests in the top section of Fig. 3C.
4. Clarified the connections and differences to a related work in Section 1.
5. Provided an explanation in Section 2.3 on why P2P cannot be computed for the body dataset.
6. Explained in Section 3.2 why CD and P2P comparisons were not conducted for nnU-Net and MedSAM2.
7. Improved methodology explanations in Section 2 by incorporating step-by-step descriptions.
8. Expanded the discussion on current limitations in Section 4.

All changes are highlighted in yellow in the revised manuscript. We provide detailed responses to each reviewer in their respective sections.

**Supporting Material:**

/attachment/2790ee9e727e1df4040c2fe32e103c8d4712c241.pdf

---

### Meta-Review · Area_Chair_gmaA · 2025-03-21

**Recommendation:** Accept (Oral)
**Confidence:** 4

**Metareview:**

This paper presents a novel mesh-prompted method for anatomical structure segmentation, leveraging a canonical mesh as a prompt to improve robustness and accuracy. The method performs mesh-to-image registration, significantly reducing ambiguity compared to point-prompted methods. Essentially, the method relies on the user selecting a template to perform a global registration, followed by network-based local registration, implicitly injecting class information and anatomy knowledge. This approach offers a new way to introduce prompts while keeping the user effort minimal.

Strengths:
- Innovative use of a new form of prompt as shape priors.
- Comprehensive experiments.

Weaknesses:
- Limited comparison with other shape-based methods.
- Computational efficiency and flexibility could be better analyzed.

One concern regarding the title: although the authors have revised it, I still feel it is not entirely appropriate. The method essentially has little to do with "mesh" itself, as it ultimately converts the input to distance maps for processing. Therefore, calling it "template" might be more suitable.

Overall, the method is original, simple, and interesting to deliver to the community. I recommend it as an oral presentation.